# Impact of Biological Agents on Postsurgical Complications in Inflammatory Bowel Disease: A Multicentre Study of Geteccu

**DOI:** 10.3390/jcm10194402

**Published:** 2021-09-26

**Authors:** María José García, Montserrat Rivero, José Miranda-Bautista, Iria Bastón-Rey, Francisco Mesonero, Eduardo Leo-Carnerero, Diego Casas-Deza, Carmen Cagigas Fernández, Albert Martin-Cardona, Ismael El Hajra, Nerea Hernández-Aretxabaleta, Isabel Pérez-Martínez, Esteban Fuentes-Valenzuela, Nuria Jiménez, Cristina Rubín de Célix, Ana Gutiérrez, Cristina Suárez Ferrer, José María Huguet, Agnes Fernández-Clotet, María González-Vivó, Blanca Del Val, Jesús Castro-Poceiro, Luigi Melcarne, Carmen Dueñas, Marta Izquierdo, David Monfort, Abdel Bouhmidi, Patricia Ramírez De la Piscina, Eva Romero, Gema Molina, Jaime Zorrilla, Cristina Calvino-Suárez, Eugenia Sánchez, Andrea Nuñez, Olivia Sierra, Beatriz Castro, Yamile Zabana, Irene González-Partida, Saioa De la Maza, Andrés Castaño, Rodrigo Nájera-Muñoz, Luis Sánchez-Guillén, Micaela Riat Castro, José Luis Rueda, José Manuel Benítez, Pedro Delgado-Guillena, Carlos Tardillo, Elena Peña, Santiago Frago-Larramona, María Carmen Rodríguez-Grau, Rocío Plaza, Pablo Pérez-Galindo, Jesús Martínez-Cadilla, Luis Menchén, Manuel Barreiro-De Acosta, Rubén Sánchez-Aldehuelo, María Dolores De la Cruz, Luis Javier Lamuela, Ignacio Marín, Laura Nieto-García, Antonio López-San Román, José Manuel Herrera, María Chaparro, Javier P. Gisbert

**Affiliations:** 1Gastroenterology Department, Hospital Universitario Marqués de Valdecilla, Universidad de Cantabria, Instituto de Investigación Sanitaria Valdecilla (IDIVAL), 37008 Santander, Spain; digrtm@humv.es (M.R.); beatriz.castros@scsalud.es (B.C.); 2Gastroenterology Department, Hospital Universitario Gregorio Marañón, Instituto de Investigación Sanitaria Gregorio Marañón (IiSGM), and Departamento de Medicina, Universidad Complutense, 28009 Madrid, Spain; pepon_miranda@hotmail.com (J.M.-B.); luisalberto.menchen@salud.madrid.org (L.M.); drnachomarin@hotmail.com (I.M.); 3Gastroenterology Department, Hospital Universitario Clínico de Santiago, 15706 Santiago de Compostela, Spain; iria.baston@gmail.com (I.B.-R.); cristina.calvino.suarez@sergas.es (C.C.-S.); manubarreiro@hotmail.com (M.B.-D.A.); laura.nieto.garcia@sergas.es (L.N.-G.); 4Gastroenterology Department, Hospital Universitario Ramón y Cajal, 28034 Madrid, Spain; pacomeso@hotmail.com (F.M.); eugenia.sanchez.rodriguez@gmail.com (E.S.); ruben.sanchez.aldehuelo@gmail.com (R.S.-A.); mibuzon@gmail.com (A.L.-S.R.); 5Gastroenterology Department, Hospital Universitario Virgen del Rocío, 41013 Sevilla, Spain; eleoc@telefonica.net (E.L.-C.); andreanuor@gmail.com (A.N.); mdcruzra@hotmail.com (M.D.D.l.C.); josemanuel.herrera@telefonica.net (J.M.H.); 6Gastroenterology Department, Hospital Universitario Miguel Servet, Instituto de Investigación Sanitaria Aragón (IISA), 50009 Zaragoza, Spain; diegocasas8@gmail.com (D.C.-D.); osierra@alumni.unav.es (O.S.); luisjalamuela@hotmail.com (L.J.L.); 7Colorectal Unit, Department of General and Digestive Surgery, Hospital Universitario Marqués de Valdecilla, 39008 Santander, Spain; carmen.cagigas@scsalud.es; 8Gastroenterology Department, Hospital Universitari Mútua Terrassa, Centro de Investigación Biomédica en Red de Enfermedades Hepáticas y Digestivas (CIBERehd), 08221 Terrassa, Spain; martincardona@gmail.com (A.M.-C.); yzabana@gmail.com (Y.Z.); 9Gastroenterology Department, Hospital Universitario Puerta de Hierro, 28220 Majadahonda, Spain; ismael.elhm@gmail.com (I.E.H.); irenegonzalezpartida@gmail.com (I.G.-P.); 10Gastroenterology Department, Hospital Universitario de Basurto, 48013 Bilbao, Spain; nerea.hernandezaretxabaleta@osakidetza.eus (N.H.-A.); saioa.delamazaortiz@osakidetza.eus (S.D.l.M.); 11Department of Gastroenterology, Hospital Universitario Central de Asturias, Instituto de Investigación Sanitaria del Principado de Asturias (ISPA), 33011 Oviedo, Spain; ipermar_79@hotmail.com (I.P.-M.); castaogarcia@gmail.com (A.C.); 12Gastroenterology Department, Hospital Universitario Río Hortega, 47012 Valladolid, Spain; efuentesv@saludcastillayleon.es (E.F.-V.); odnaj@hotmail.com (R.N.-M.); 13Gastroenterology Department, Hospital General Universitario de Elche, 03203 Alicante, Spain; nujigar@hotmail.com; 14Gastroenterology Department, Hospital Universitario de La Princesa, Instituto de Investigación Sanitaria Princesa (IIS-IP), Universidad Autónoma de Madrid (UAM), Centro de Investigación Biomédica en Red de Enfermedades Hepáticas y Digestivas (CIBERehd), 28006 Madrid, Spain; cristina.rubin.92@hotmail.com (C.R.d.C.); micariat4@gmail.com (M.R.C.); mariachs2005@gmail.com (M.C.); javier.p.gisbert@gmail.com (J.P.G.); 15Gastroenterology Department, Hospital General de Alicante, Centro de Investigación Biomédica en Red de Enfermedades Hepáticas y Digestivas (CIBERehd), Instituto de Investigación Sanitaria y Biomédica de Alicante (ISABIAL), 03010 Alicante, Spain; gutierrez_anacas@gva.es; 16Gastroenterology Department, Hospital Universitario La Paz, 28046 Madrid, Spain; cristinajsuarezferrer@gmail.com (C.S.F.); ruedagarcia.joseluis@gmail.com (J.L.R.); 17Gastroenterology Department, Hospital General Universitario de Valencia, 46014 Valencia, Spain; josemahuguet@gmail.com; 18Gastroenterology Department, Hospital Clinic of Barcelona, 08036 Barcelona, Spain; agfernandez@clinic.cat; 19Gastroenterology Department, Hospital del Mar, 08003 Barcelona, Spain; mariagvivo@gmail.com; 20Gastroenterology Department, Hospital Rafael Méndez, 30817 Lorca, Spain; blanca.dvo@gmail.com; 21Gastroenterology Department, Hospital Sant Joan Despí-Moisès Broggi, 08970 Barcelona, Spain; jesus.castropoceiro@sanitatintegral.org; 22Gastroenterology Department, Hospital Universitari Parc Taulí, Sabadell, Centro de Investigación Biomédica en Red de Enfermedades Hepáticas y Digestivas (CIBERehd), 08208 Barcelona, Spain; lmelcarne@outlook.com; 23Gastroenterology Department, Hospital Universitario de Cáceres, 10003 Cáceres, Spain; cdsadornil@gmail.com; 24Gastroenterology Department, Hospital Universitario de Cabueñes, 33203 Gijón, Spain; martaizquierdoromero@gmail.com; 25Gastroenterology Department, Consorcio Sanitario de Terrasa, 08227 Barcelona, Spain; dmonfort@cst.cat; 26Gastroenterology Department, Hospital de Santa Bárbara, 13500 Puertollano, Spain; bumidi@hotmail.com; 27Gastroenterology Department, Hospital Universitario Vitoria-Gastéiz, 01002 Vitoria, Spain; patri_rami@hotmail.com; 28Gastroenterology Department, Hospital Clínico Universitario de Valencia, 46010 Valencia, Spain; romeroglez.eva@gmail.com; 29Gastroenterology Department, Hospital Arquitecto Marcide, 15405 Ferrol, Spain; gma.torde@hotmail.com; 30Department of Colorectal and Gastrointestinal Surgery, Hospital Universitario Gregorio Marañón, 28009 Madrid, Spain; jaime.zorrilla@salud.madrid.org; 31Department of Colorectal and Gastrointestinal Surgery, Hospital General Universitario de Elche, 03203 Alicante, Spain; drsanchezguillen@gmail.com; 32Gastroenterology Department, Hospital Reina Sofía, IMIBIC, 14004 Córdoba, Spain; jmbeni83@hotmail.com; 33Gastroenterology Department, Hospital General de Granollers, 08042 Granollers, Spain; pgdg20@gmail.com; 34Gastroenterology Department, Hospital Nuestra Señora de la Candelaria, 38010 Tenerife, Spain; cartardillo@gmail.com; 35Gastroenterology Department, Hospital Royo Villanova, 50007 Zaragoza, Spain; epenagon80@yahoo.es; 36Gastroenterology Department, Complejo Hospitalario de Soria, 42005 Soria, Spain; santifrago@gmail.com; 37Gastroenterology Department, Hospital Universitario de Henares, 28002 Coslada, Spain; mc.r.grau@gmail.com; 38Gastroenterology Department, Hospital Universitario Infanta Leonor, Vallecas, 28031 Madrid, Spain; rocio_plaza@yahoo.es; 39Gastroenterology Department, Complejo Hospitalario Universitario de Pontevedra, 36071 Pontevedra, Spain; perez.galindo.pablo@gmail.com; 40Gastroenterology Department, Hospital Álvaro Cunqueiro de Vigo, 36312 Vigo, Spain; jmcadilla@hotmail.com

**Keywords:** inflammatory bowel disease, Crohn’s disease, ulcerative colitis, anti-TNF, ustekinumab, vedolizumab, postoperative complications, surgery, preoperative therapy

## Abstract

Background: The impact of biologics on the risk of postoperative complications (PC) in inflammatory bowel disease (IBD) is still an ongoing debate. This lack of evidence is more relevant for ustekinumab and vedolizumab. Aims: To evaluate the impact of biologics on the risk of PC. Methods: A retrospective study was performed in 37 centres. Patients treated with biologics within 12 weeks before surgery were considered “exposed”. The impact of the exposure on the risk of 30-day PC and the risk of infections was assessed by logistic regression and propensity score-matched analysis. Results: A total of 1535 surgeries were performed on 1370 patients. Of them, 711 surgeries were conducted in the exposed cohort (584 anti-TNF, 58 vedolizumab and 69 ustekinumab). In the multivariate analysis, male gender (OR: 1.5; 95% CI: 1.2–2.0), urgent surgery (OR: 1.6; 95% CI: 1.2–2.2), laparotomy approach (OR: 1.5; 95% CI: 1.1–1.9) and severe anaemia (OR: 1.8; 95% CI: 1.3–2.6) had higher risk of PC, while academic hospitals had significantly lower risk. Exposure to biologics (either anti-TNF, vedolizumab or ustekinumab) did not increase the risk of PC (OR: 1.2; 95% CI: 0.97–1.58), although it could be a risk factor for postoperative infections (OR 1.5; 95% CI: 1.03–2.27). Conclusions: Preoperative administration of biologics does not seem to be a risk factor for overall PC, although it may be so for postoperative infections.

## 1. Introduction

Inflammatory bowel disease (IBD) management completely changed after the approval by the European Medicines Agency (EMA) of the first anti-tumor necrosis factor (TNF) in 1999 [1]. Since then, biologics have increased the therapeutic armamentarium previously based on corticosteroids, immunomodulators and surgery. The development of these therapies exerted a positive impact on the natural history of IBD and an improvement in the control of inflammation [2]. However, only a proportion of patients respond to medical therapy and surgery still has a fundamental role in the management of IBD [3]. For this reason, 50% of the patients affected by Crohn’s disease (CD) and 10–20% of ulcerative colitis (UC) patients require surgery within 10 years after diagnosis [4,5]. Furthermore, 15–20% of those surgeries suffer from postoperative complications, thus preventing these side effects is highly relevant [6,7].

Several risk factors related to postoperative complications have been identified, such as preoperative corticosteroid administration, malnutrition, hypoalbuminemia or other factors associated to the surgical procedure, such as the experience of the surgeon or the surgery approach [8,9,10]. Regarding preoperative treatment, the preoperative administration of thiopurines or methotrexate does not seem to be associated with a higher risk of postoperative complications [11]. 

Several studies have evaluated the risk of postoperative complications in patients treated with biologics, mainly anti-TNF, obtaining conflicting results [12,13]. Furthermore, safety data about more recently approved biologics, such as vedolizumab and ustekinumab, in this setting are limited [14,15]. Therefore, the safety of preoperative biological therapy within the preoperative period remains unclear. A high proportion of patients who undergo surgery are using biological agents and, therefore, knowing whether this treatment poses a higher risk of complications is of utmost importance in determining whether to schedule surgery. 

Therefore, our aim was to evaluate the impact of preoperative biological therapy (not only anti-TNF but also vedolizumab and ustekinumab) on the risk of postsurgical complications (mainly focused on infections). In addition, we aimed to identify clinical characteristics, surgical procedures and any treatment administered during the preoperative period that might impact on patients’ outcomes. Thus, our study will contribute to improve the knowledge of the safety of these treatments during the postoperative period. 

## 2. Materials and Methods

### 2.1. Study Design and Population

We designed a multicentre retrospective study of patients who required abdominal surgery as treatment for IBD. Patients above 18 years old who required surgery between 1 January 2009 and 31 December 2019 were included. This period was chosen after considering the approval date of IBD biological therapy to establish a homogeneous management of these diseases in Spain. Pregnant women, patients on immunosuppressants for diseases other than IBD, patients on biologicals for diseases other than IBD or patients who underwent surgeries for perianal disease were excluded. In order to establish the risk of these patients, we compared two groups: the exposed cohort, which was comprised of patients whose last dose of biological therapy had been administered at any point during 12 weeks before the date of surgery, and the non-exposed cohort, which was comprised of patients who had not been subjected to any biological therapy in the same period. Once the surgeries were assigned to each group, the clinical characteristics of both categories were studied and their differences concerning clinical features, biochemical parameters, preoperative treatments and surgical procedures were analysed. Surgeries with and without complications were compared according to the presence of biological therapy during the preoperative period. Postsurgical infections were also separately analysed because they are especially relevant complications.

The study was conducted by the Young Group of the Spanish Working Group of Crohn’s disease and Ulcerative Colitis (GETECCU). The study was carried out in accordance to the European General Data Protection Regulation (GDPR) 2016/679 and the Spanish Data Protection Organic Law 3/2018. The protocol was approved by the Research Ethics Committees of each centre and the Spanish Agency of Medicines and Medical Devices (code MJG-VED-2019-01). 

### 2.2. Data Collection

All patients diagnosed with IBD were distributed into three categories, namely CD, UC and IBD-unclassified, according to the recommendations set by the European Crohn and Colitis Organisation (ECCO). The location and the severity of IBD at the time of surgery was recorded according to the Montreal Classification. Data collection included demographic characteristics such as sex, date of birth, IBD diagnosis date, smoking habit at the time of surgery and anthropometric measurements. The Harvey-Bradshaw index and partial Mayo score as well as laboratory parameters including nutritional status were recorded two weeks before the date of surgery. The parameter closer to the date of surgery was chosen when more than one were found in the medical records. Data of corticosteroid, immunomodulator administration previous to the date of surgery were also collected. The biologic agents included during the preoperatory period were infliximab, adalimumab, golimumab, vedolizumab and ustekinumab. Regarding the surgical procedure, indication, whether surgery was urgent or elective, type of surgery, postoperative complications, length of hospital stay, 30-day hospital readmission, 30-day surgical requirements to control complications and 90-day death rate were recorded. Clavien-Dindo classification was used to assess the severity of complications [16]. The centres involved in the study were categorized in 5 levels, according to parameters such as number of hospital beds, local population assigned, the existence of university teaching and available diagnostic tests such as on-site nuclear or radiological techniques, with 5 being the maximum score for these parameters. 

Study data were collected by an electronic data capture tool (Research Electronic Data Capture (REDCap), which is hosted by Asociación Española de Gastroenterología (AEG; www.aegastro.es) [17]. AEG provided this service free of charge, with the sole aim of promoting independent investigator-driven research. REDCap is a secure, web-based application designed that supports data capture for research studies and provides an intuitive interface for validated data entry, audit trails for tracking data manipulation and export procedures, automated export procedures for seamless data downloads to common statistical packages and procedures for importing data from external sources.

### 2.3. Definitions

−Postoperative complications: the presence of superficial wound infection, intraabdominal infection, urinary tract infection, bacteraemia, respiratory infection, fever above 38 °C of unknown origin, anastomosis leak, mechanical obstruction, postoperative ileus, bleeding, thrombosis, fistula or evisceration during the 30 days after the date of surgery.−Anaemia: haemoglobin level under 12 g/dL for women and under 13 g/dL for men at any point during the two weeks prior to surgery [18]. Severe anaemia was considered when haemoglobin level was under 10 g/dL regardless of the sex [19]. −Low albumin levels: albumin levels lower than 3 g/dL at any point during the two weeks before the date of surgery [20]. −Low cholesterol levels: serum cholesterol level below 160 mg/dL at any point during the two weeks prior to surgery [10]. −Smoking habit: current smokers included individuals who actively smoked more than seven cigarettes per week, former smokers included individuals who quit smoking more than six months ago and non-smokers included those patients who had never smoked before [21].−Nutritional risk: a weight loss >10% within six months or body mass index (BMI) <18.5 kg/m^2^ [22].

### 2.4. Statistical Analysis

Quantitative variables are expressed as mean and standard deviation or median and interquartile range, depending on whether they have a normal distribution or not. Qualitative variables are expressed as percentages and 95% confidence intervals (CI). Chi-square test or the Fisher exact test were used to compare qualitative variables, while differences of quantitative variables between the two groups were analysed by the Student *t*-test or the Wilcoxon-rank sum test depending on data distribution. A significant result was considered when the *p*-value was ≤0.05 for the overall comparison of both groups (exposed to biological therapy or non-exposed to these drugs). The analysis was performed separately for each variable. Afterwards, a multivariate analysis through binary logistic regression was carried out to compare the risk of every variable with respect to the risk of postoperative complications as well as the risk of postoperative infections. Two models were evaluated: the first model included the perioperative administration of biological therapy as a binary variable, while the second model evaluated the biological therapy in 3 categories (anti-TNF, ustekinumab and vedolizumab). All the variables with a univariate *p* < 0.20 and those that were clinically relevant were evaluated in the multivariate analysis as independent variables while the presence of postoperative complications was considered as the dependent variable. All statistical analyses were performed with STATA Statistical Software: Release 14. StataCorp LP. 

A sensitivity analysis through propensity score was performed to evaluate baseline variables that could have an influence on the results. The variables included in the propensity score were those clinically or statistically significant through logistic regression, biological exposure being the dependent variable. The confounding factors included were carefully discussed, evaluated and selected before the data analysis. Surgeries were matched one-to-one through the genetic matching method and the covariates were balanced for both groups [23]. To evaluate the balance of each variable, a graphic representing the means of each covariate compared to the estimated propensity score was made after matching by exposure.

## 3. Results

### 3.1. Patient Population

A total of 1535 IBD surgeries in 1370 patients from 37 hospitals were performed. Baseline characteristics of both groups are detailed in Table 1. Overall, in 584 surgery patients had been exposed to anti-TNF before surgery, 58 to vedolizumab and 69 to ustekinumab. In thirty-five percent of the surgeries there was no previous exposure to biological therapy at any point during the disease course, while patients had been treated with one biological treatment in 40% of the surgeries, with two biological treatments in 16.9% and with three or more in 8.3% of the surgeries. Regarding the type of intervention, small bowel surgery was the most frequent in 48.8% of the cases, followed by colonic surgery (26.6%), ileocolonic surgery (19.0%) and restorative surgery (5.6%).

### 3.2. Postoperative Complications

Postoperative complications were observed in 35.6% (95% CI: 33.2–38.1, *n* = 547) of the surgeries; 37.6% (95% CI: 34.0–41.2) in the exposed cohort and 34.0% (95% CI: 30.7–37.3) in the non-exposed cohort (*p* = 0.15). The most frequently found postoperative complications were infections, which occurred in 48.0% of the cases, followed by anastomosis leak in 15.6%, postoperative ileus in 12.4% and bleeding in 12.2% of the overall complications. Of surgeries with complications, 83.6% (*n* = 457) had one complication, 13.7% (*n* = 75) two complications, 2.2% (*n* = 12) three complications, and 0.6% (*n* = 3) more than three complications. According to exposure, 20.8% (*n* = 148) of postoperative infections were assigned to the exposed cohort and 19.3% (*n* = 159) to the non-exposed (*p* = 0.5). Using the Clavien-Dindo classification we grouped the complications according to severity levels; 55.2% (*n* = 302) of the cases required pharmacologic treatment without surgery, 35.1% (*n* = 192) needed endoscopic, radiological or surgical intervention and 9.7% (*n* = 53) of the surgeries presented a life-threating complication. Hospital readmission within 30 days after hospital discharge was needed in 7.2% (*n* = 110) of the patients and 1.9% (*n* = 29) required a new surgery. The 90-day mortality rate reached 0.7% (*n* = 11) of the surgeries. No significant differences in complication rates, Clavien-Dindo classification, type of complication, hospital readmission or the need for a new surgery were observed according to treatment exposure. Detailed data of this analysis is presented in Table 2. 

### 3.3. Postoperative Complications According to Exposure 

When we grouped the cohort according to the exposure, 46.3% (95% CI: 43.8–48.9, *n* = 711) had received a biological treatment during the preoperative period and 53.7% (95% CI: 51.1–56.2, *n* = 824) of the surgeries had not. We found that the exposed cohort was composed of younger patients, with lower median age at the time of IBD surgery, higher proportion of stricturing behaviour, perianal disease and extraintestinal manifestations in comparison to the non-exposed cohort. Furthermore, more hospital admissions within three months before the date of surgery were registered in the exposed cohort (43.7% vs. 32.2%, *p* = 0.001), as well as higher Mayo scores (6.9 points vs. 4.2 points, *p* ≤ 0.0001).

According to anthropometric and laboratory parameters, more weight loss within six months prior to surgery and lower levels of haemoglobin were observed in the exposed cohort resulting in an increased use of blood transfusions and intravenous iron in that group. Furthermore, more nutritional support was administered in that cohort, although no differences in cholesterol, albumin and prealbumin levels were observed between both groups (Table 1). 

### 3.4. Predictive Factors Associated with the Appearance of Postoperative Complications 

The factors associated with patients experiencing more postoperative complications as determined in the univariate analysis were male gender, age over 40 years at the time of surgery, a diagnosis of UC, severe anaemia, corticosteroid use, higher levels of C-reactive protein (CRP) and nutritional parameters such as low serum cholesterol and albumin levels during the preoperative period (Table 3). Surgical techniques were also analysed, finding higher risk in emergency surgeries, colonic surgeries, pouch surgeries and in those performed by laparotomy (Table 3 and Table 4). 

In the multivariate analysis, the factors that posed a risk for surgical complications were male gender, requirement of urgent surgery, need for laparotomy approach and haemoglobin levels under 10 gr/dL during the preoperative period. In contrast, being operated in centres whose category was 5 led to a reduction in the risk of postoperative complications (Table 5). Regarding the preoperative treatment for IBD, biological therapy was not associated with postoperative complications in the multivariate analysis (OR 1.24; 95% CI: 0.97–1.58). 

Focusing on postoperative infections, the multivariate analysis showed that the patients that received biological therapy during the preoperative period were at increased risk of developing postoperative infections, with borderline statistical significance (OR 1.50; 95% CI: 1.03–2.17). Moreover, this result was confirmed in the propensity score, which showed a significant result for postoperative infections in patients exposed to biological therapy during the preoperative period. Other factors that influenced the risk of postoperative infections were high levels of CRP, hypoalbuminaemia, and the requirement of laparotomy (Table 5).

### 3.5. Type of Biological Therapy during the Preoperative Period and Its Impact on Postoperative Complications

As previously mentioned, in the multivariate analysis the use of biological therapy during the preoperative period was not associated with suffering from overall postoperative complications. Furthermore, biological intensification during the preoperative period did not influence postsurgical complications (*p* = 0.7). The groups defined according to prior biological treatment were no biological therapy (584 surgeries), anti-TNF (261 exposed to adalimumab and 323 exposed to infliximab), vedolizumab (58) and ustekinumab (69). Regarding the type of IBD, for UC 101 cases had received anti-TNF, 28 vedolizumab, three ustekinumab and 149 no biological therapy, while for CD 477 had received anti-TNF, 30 vedolizumab, 66 ustekinumab and 665 no biological therapy. Results of the univariate analysis of association between preoperative biological treatment and postsurgical complication are shown in Figure 1 for IBD, UC and CD. In the multivariate analysis, no specific treatment was associated with postoperative complications or infections. Regarding other therapies, no statistically significant differences were found for corticosteroids or immunomodulators during the preoperative period. 

### 3.6. Sensitivity Analysis

The estimation of the exposure to biological therapy during the preoperative period and its influence on postoperative complications and postoperative infections was confirmed in the propensity score matching analysis estimated with the following variables: mean age at surgery, age at IBD onset, average duration of IBD until surgery, extraintestinal manifestations, smoking habit, perianal disease, prior IBD surgery, need for nutritional support, haemoglobin level, and the need for transfusion. In the matched cohort, all standardised differences were below 10%. The means of each covariate compared to the estimated propensity score were represented in graphs, finding no significant differences (Figure 1, Appendix A). In the matched cohort ORs were 1.4 (95% CI: 0.85–2.33) for postoperative complications and 2.33 (95% CI: 1.12–4.07) for postoperative infections. 

## 4. Discussion

To our knowledge, this is the largest cohort study that has evaluated the safety of preoperative anti-TNF, vedolizumab or ustekinumab treatments in IBD patients. Our results demonstrate that preoperative administration of biologics is not associated with overall postoperative complications in IBD patients, although it may be a risk factor for postoperative infections. In the sensitivity analysis, the risk of postoperative complications was similar in the non-matched and the matched cohort so the differences in clinical characteristics do not affect the results of the study.

Although multiple studies have evaluated the risk of biological therapy during the preoperative term, its effect is still under debate. Similar incidences of postoperative complications in patients with or without this therapy was observed in our cohort. The preliminary data of several meta-analyses showed a higher risk of complications in IBD patients treated with anti-TNF, especially in those with CD [24,25]. In contrast to these data, the administration of preoperative infliximab was not related to the appearance of early postoperative complications in recent meta-analyses for CD [26,27]. Furthermore, the only two studies that evaluated this effect prospectively showed that neither anti-TNF administration nor anti-TNF drug levels during the preoperative period was associated with postoperative complications in IBD; therefore, the complete withdrawal of biological therapy during the preoperative period is not necessary to reduce the frequency of postoperative complications [28,29].

Data on recently approved treatments and their implications on the risk of postoperative complications are limited, as comparative studies have only been published since 2017. Our study is the first one analysing anti-TNF, vedolizumab, and ustekinumab, using a cohort of IBD patients with no preoperative biological therapy as control. In our study, no statistical differences were observed in the multivariate analysis between the different types of biological therapy. Only one study compared these treatments, exclusively for CD, and it had similar results [30]. Regarding vedolizumab, previous publications reported that this treatment was not an independent risk factor for developing postoperative complications compared to anti-TNF and ustekinumab [31,32]. However, more postoperative ileus was found after vedolizumab administration during the preoperative period compared to anti-TNF and no biological therapy [33].

Our cohort is also the largest reported to date analysing the preoperative administration of ustekinumab and its effect during the postoperative period. This therapy was recently approved for UC; accordingly, no information concerning its effect on this disease has ever been published. In our cohort only three UC patients were treated with ustekinumab, hence no conclusions could be established. Only two studies evaluated the association between previous ustekinumab administration and complications in CD [34,35]. Based on these preliminary data and according to previous publications, withdrawal of ustekinumab or vedolizumab before a surgical procedure does not seem to be required in routine practice to avoid postoperative complications. 

Regarding postoperative infections, the exposure to biological therapy seemed to be an independent risk factor in our patient cohort, although the results only reached borderline statistical significance. A recent meta-analysis revealed a slightly higher incidence of infections in patients under anti-TNF therapy, although this effect was not observed for vedolizumab [36,37]. Discordance of results for anti-TNF agents could be influenced by therapeutic plasma concentrations of anti-TNF at the time of surgery [38]. Regarding vedolizumab and infection complications, only one study linked the preoperative administration of anti-integrins to a higher proportion of superficial wound infections, whereas no association was found in other studies [39,40,41]. Similarly, ustekinumab administration is not a risk factor for postoperative infections, even though its use was associated with intraabdominal sepsis after surgery in a single-centre study [34,42,43]. It is worth mentioning that, according to other studies, calcineurin inhibitors, thiopurines or methotrexate do not pose a risk for postoperative complications or infections [44,45].

Although one-third of all the patients in the current study had received corticosteroids before surgery, their effect was only detected in the univariate analysis, whereas hypoalbuminaemia was an independent risk factor for suffering from postoperative infections in the multivariate analysis. Corticosteroids are known to be one of the most important factors affecting the incidence of postoperative complications through their effect on wound healing and the bursting pressure of the healing [8,46]. Albumin and nutritional status are also essential factors to evaluate during preoperatory management, despite the fact that a higher risk of complications has been observed in those patients with mixed or exclusive parenteral nutrition [47]. Of note, corticosteroids and hypoalbuminaemia are intimately associated with other factors involved in postoperative complications such as anaemia, the temporality of surgery or the surgical approach [48,49,50]. Regarding anaemia, only one study analysed the association between its severity and the risk of complications in IBD [51]. We report that suffering from anaemia before surgery is also a significant risk factor for postoperative complications. Its influence has been also recognized in other diseases such as colorectal cancer, hence the preoperative management of this condition is recommended in IBD [52,53]. Analysing the temporality of the surgery, we observed that urgent surgeries increased the rate of complications compared to elective ones; and the use of the laparotomy approach during surgery also increased complications, as described in previous reports [54,55,56,57]. Moreover, infections were linked to high CRP levels in our cohort [58]. For this reason, a balance of risk and benefit has to be assessed, trying to optimize the preoperative status of the patient by a multidisciplinary team, avoiding surgery delays, monitoring clinical condition and performing the surgery in referral centres when possible [59,60].

One of the limitations of our study is retrospective data collection. Also, the postoperative events included as complications depend on their definition in each study, thus their incidence could differ, thereby affecting the results between studies. However, the Clavien-Dindo classification, which has been used as an outcome in previous reports, was used to avoid this limitation by making an effort at standardising our data [61]. Nevertheless, neither patient comorbidity nor the risk associated with the anaesthetic procedure was collected. Another important aspect is the recent approval of vedolizumab or ustekinumab, which limits the number of patients treated with those drugs compared to anti-TNF therapy. On the other hand, a strength of our study is the application of the genetic matched score. The use of this method to compare cohorts improved the quality of our results in comparison to previous studies that did not utilize this analysis. Furthermore, our study is one of the largest cohorts for IBD patients encompassing both different hospital categories and various types of biological therapy. For that reason, our results show real-world postoperative complications and not only those from referral centres. 

## 5. Conclusions

In conclusion, the preoperative administration of biological therapy does not seem to increase the risk for overall postoperative complications in IBD, although it may be a specific risk factor for postoperative infections. The need for urgent surgery, the laparotomy approach, severe anaemia as well as the type of hospital have to be considered as risk factors for developing postoperative complications. Finally, hypoalbuminaemia, the laparotomy approach and higher CPR levels increase the risk of developing postoperative infections.

## Figures and Tables

**Figure 1 jcm-10-04402-f001:**
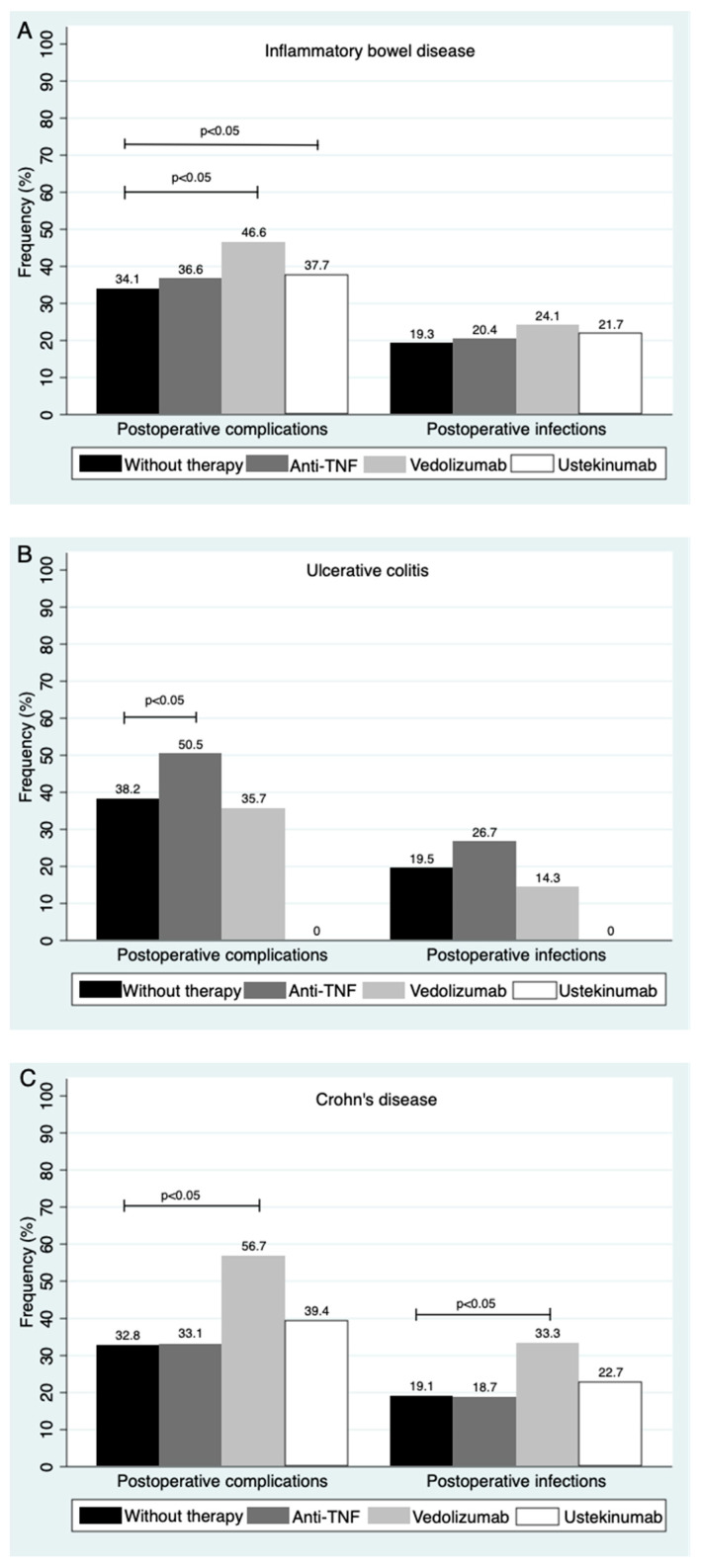
Effect of biological treatment during the preoperative period on frequency of postoperative complications and infections by Chi-square test. (**A**), Inflammatory bowel disease. (**B**), Ulcerative colitis. (**C**), Crohn’s disease. Statistically significant differences (*p* ≤ 0.05) are indicated in the graphic.

**Table 1 jcm-10-04402-t001:** Clinical characteristics of the surgeries according to prior exposition to biological therapy. *p*-values were calculated by Chi-square test, *t*-test or Wilcoxon-rank sum.

	Exposed Cohort (*n* = 711)	Non-Exposed Cohort (*n* = 824)	*p*-Value
Gender: male	51.5 (363)	53.8 (443)	0.3
Median age at surgery (years) (mean, SD)	43.57 (13.48)	46.26 (15.36)	<0.001 *
Median age at IBD onset (years) (mean, SD)	33.43 (13.74)	37.40 (16.03)	<0.001 *
Mean duration of IBD until surgery (years) (mean, SD)	10.13 (8.56)	8.85 (9.05)	<0.05 *
Smoking habit (%, *n*)			<0.05 *
- Current smokers	25.2 (170)	31.6 (242)	
- Former smokers	25.2 (170)	18.8 (144)	
- Non smokers	49.7 (336)	49.5 (379)	
Type of disease (%, *n*)			0.76
- Ulcerative colitis	18.76(132)	18.1 (149)	
- Crohn’s disease	80.6 (573)	80.7 (665)	
- IBD-unclassified	0.8 (6)	1.2 (10)	
Location of IBD (%, *n*)			
- Ulcerative proctitis (UC)	3.6 (5)	0.6 (1)	0.08
- Left-side colitis (UC)	23.2 (32)	18.2 (29)	
-Extensive colitis (UC)	73.2 (101)	81.1 (129)	
- Ileum (CD)	49.2 (282)	53.4 (355)	0.13
- Colon (CD)	5.8 (33)	7.1 (47)	
- Ileocolonic (CD)	45.0 (258)	39.6 (263)	
- Upper disease (CD)	10.8 (62)	7.5 (50)	
Behaviour of CD at surgery (%, *n*)			<0.05 *
- Inflammatory	13.3 (76)	16.5 (110)	
- Stricturing	56.5 (324)	46.3 (308)	
- Penetrating	30.2 (173)	37.1 (247)	
Perianal disease (yes) (%, *n*)	24.4 (140)	17.1 (14)	<0.05 *
Extraintestinal manifestations (yes) (%, *n*)	21.9 (156)	15.7 (129)	<0.05 *
Prior surgery for IBD (yes) (%, *n*)	31.1 (221)	35.8 (295)	0.05
Hospital admission within 3 months prior to surgery (yes) (%, *n*)	43.7 (310)	32.2 (265)	<0.001 *
Partial Mayo Score (mean, SD)	6.89 (2.27)	4.2 (3.04)	<0.001 *
Harvey-Bradshaw Index (mean, SD)	6.56 (3.59)	6.38 (3.28)	0.47
Weight at surgery (kg) (mean, SD)	64.18 (14.23)	65.99 (14.49)	0.08
Weight loss between 6 months and 2 weeks prior to surgery (kg) (mean, SD)	4.52 (8.73)	3.09 (7.18)	<0.05 *
BMI at surgery (mean, SD)	22.81 (4.53)	23.31 (4.48)	0.13
Haemoglobin (gr/dL) (mean, SD)	12.19 (1.98)	12.63 (2.11)	<0.001 *
Lymphocyte count (/mL) (mean, SD)	1895.51 (1096.27)	1702.5 (1013.08)	<0.001 *
C-reactive protein (mg/dL) (mean, SD)	4.53 (13.61)	5.05 (8.43)	0.47
Cholesterol (mg/dL) (mean, SD)	149.60 (43.40)	153.66 (43.52)	0.23
Prealbumin (mg/dL) (mean, SD)	21.84 (9.20)	21.41 (10.35)	0.76
Albumin (mg/dL) (mean, SD)	3.52 (0.70)	3.59 (0.78)	0.14
Malnutrition (yes) (%, *n*)	43.7 (151)	37.53 (158)	0.08
Blood transfusion (yes) (%, *n*)	13.5 (96)	6.9 (57)	<0.001 *
Intravenous iron treatment (yes) (%, *n*)	22.9 (163)	13.0 (107)	<0.001 *
Type of preoperative nutrition support (%, *n*)			<0.001 *
- No supplementary nutrition	61.6 (438)	77.3 (637)	
- Enteral	20.4 (145)	11.5 (95)	
- Parenteral	9.3 (66)	8.0 (66)	
- Enteral and parenteral	8.7 (62)	3.2 (26)	
Corticosteroids (yes) (%, *n*)	38.1 (271)	28.1 (231)	<0.001 *
Immunomodulators (yes) (%, *n*)	43.7 (311)	24.4 (201)	<0.001 *

SD = standard deviation; IBD = inflammatory bowel disease, UC = ulcerative colitis; CD = Crohn’s disease; BMI = body mass index; * = statistical significance

**Table 2 jcm-10-04402-t002:** Effect of biological treatment on the incidence of postoperative complications calculated by Chi-square test.

	Exposed Cohort	Non-Exposed Cohort	*p*-Value
Overall complications (%, *n*)	37.6 (267)	34.0 (280)	0.15
Superficial wound infection (%, *n*)	7.7 (55)	7.5 (62)	0.8
Intraabdominal infection (%, *n*)	10.4 (74)	9.3 (77)	0.5
Other infections (%, *n*)	3.4 (24)	3.9 (32)	0.5
Anastomosis leak (%, *n*)	7.0 (50)	6.9 (57)	0.9
Bowel obstruction (%, *n*)	2.0 (14)	1.2 (10)	0.2
Postoperative ileus (%, *n*)	6.5 (46)	4.6 (38)	0.1
Bleeding (%, *n*)	5.2 (37)	5.2 (43)	0.9
Thrombosis (%, *n*)	0.4 (3)	0.7 (6)	0.4
Fistula (%, *n*)	0.8 (6)	1.0 (8)	0.8
Evisceration (%, *n*)	0.1 (1)	0.73 (6)	0.09

**Table 3 jcm-10-04402-t003:** Clinical and therapeutic features related to the presence of postoperative complications. *p*-values were calculated by Chi-square test, *t*-test or Wilcoxon-rank sum.

		Postoperative Complications (547 Surgeries)	Non-Complications (988 Surgeries)	*p*-Value
Gender (%, *n*)	Men	59.4 (325)	48.7 (481)	<0.001 *
Age at surgery (years) (%, *n*)	Younger than 40	34.9 (191)	44.3 (438)	<0.001 *
	Between 40 and 60	48.0 (262)	40.7 (402)	
	Older than 60	17.2 (94)	15.0 (148)	
Smoking habit (%, *n*)	Current smoker	27.8 (141)	29.0 (271)	0.84
	Former smoker	22.45(114)	21.4 (200)	
	Non smoker	49.7 (252)	49.6 (463)	
Type of disease (%, *n*)	Ulcerative colitis	21.6 (118)	16.5 (163)	<0.05 *
	Crohn’s disease	76.6 (419)	82.9 (819)	
	IBD-unclassified	1.8 (10)	0.6 (6)	
Location at surgery (%, *n*)	Extensive colitis	83.0 (98)	74.2 (122)	0.21
	Left-side colitis	15.2 (18)	23.3 (38)	
	Proctitis	1.7 (2)	2.5 (4)	
	Ileal (L1)	44.4 (186)	55.1 (451)	<0.001 *
	Colic (L2)	47.3 (198)	39.4 (323)	
	Ileocolic (L3)	8.4 (35)	5.5 (45)	
	Upper (L4)	8.1 (34)	9.5 (78)	
Behaviour (only CD) (%, *n*)	Inflammatory	18.1 (76)	13.4 (110)	0.07
	Stricturing	48.0 (201)	52.6 (431)	
	Penetrating	33.9 (142)	33.9 (278)	
Perianal disease (%, *n*)	Yes	19.9 (109)	16.7 (165)	0.12
	No	80.1 (438)	83.3 (823)	
Prior IBD surgery (%, *n*)	Yes	35.3 (193)	32.7 (323)	0.3
	No	64.7 (355)	67.3 (665)	
Prior non-IBD surgery (%, *n*)	Yes	18.1 (99)	17.5 (173)	0.77
	No	81.9 (448)	82.5 (815)	
Extraintestinal manifestations (%, *n*)	Yes	19.9 (109)	17.8 (176)	0.3
	No	80.0 (438)	82.2 (812)	
Severe anaemia (%, *n*)	Yes	17.7 (81)	10.0 (81)	<0.001 *
	No	82.3 (376)	90.0 (732)	
Low albumin levels (%, *n*)	Yes	28.7 (93)	14.9 (84)	<0.001 *
	No	71.3 (231)	85.1 (479)	
Low cholesterol levels (%, *n*)	Yes	64.9 (163)	55.8 (235)	<0.05 *
	No	35.1 (88)	44.2. (186)	
Intravenous iron treatment (%, *n*)	Yes	21.4 (117)	15.5 (153)	<0.05 *
	No	78.6 (430)	84.5 (835)	
Blood transfusion (%, *n*)	Yes	15.2 (83)	7.1 (70)	<0.001 *
	No	84.8 (464)	92.9 (918)	
Type of nutritional support (%, *n*)	Enteral	41.4 (72)	58.7 (168)	<0.001 *
	Parenteral	33.3 (58)	25.9 (74)	
	Enteral and parenteral	25.3 (44)	15.4 (44)	
Glucocorticoids (%, *n*)	Yes	36.3 (198)	30.8 (304)	<0.05 *
	No	63.7 (347)	69.2 (683)	
Immunomodulator therapy (%, *n*)	Yes	32.9 (180)	33.6 (332)	0.78
	No	67.1 (367)	66.4 (656)	
Biological therapy (%, *n*)	Yes	48.8 (267)	44.9 (444)	0.15
	No	51.2 (280)	55.1 (544)	
Temporality of surgery (%, *n*)	Urgent	23.8 (130)	15.3 (151)	<0.001 *
	Elective	76.2 (417)	84.7 (837)	
Surgical approach (%, *n*)	Laparotomy	73.5 (402)	67.3 (665)	<0.05 *
	Laparoscopy	26.5 (145)	32.7 (323)	
Hospital level	2nd, 3rd or 4st category	42.7 (234)	36.6 (362)	<0.05 *
	5th Category	57.2 (313)	63.4 (626)	

IBD = inflammatory bowel disease; CD = Crohn’s disease; * = statistical significance

**Table 4 jcm-10-04402-t004:** Univariate analysis of surgical procedures as risk factors for postsurgical complications calculated by logistic regression.

	Unadjusted Odds Ratio	95% Confidence Interval
Ileocecal resection	0.58	0.47–0.73
Bowel resection	0.90	0.63–1.27
Strictureplasty	1.68	0.70–4.03
Partial colonic resection	1.45	1.03–2.04
Subtotal colectomy	1.62	1.56–2.30
Total colectomy	1.72	1.06–2.79
Proctectomy	1.93	1.29–2.90
Pouch surgery	1.69	1.05–2.70

**Table 5 jcm-10-04402-t005:** Risk factors for postoperative complications and infections in the multivariate analysis calculated by logistic regression.

**Postoperative Complications**	**Adjusted Odds Ratio**	**95% Confidence Interval**
Exposure to biological therapy	1.24	0.97–1.58
Gender: male	1.54	1.21–1.95
Severe anaemia	1.83	1.30–2.57
Urgent surgery	1.61	1.21–2.16
Surgical approach: laparotomy	1.45	1.11–1.90
Hospital level: 5th category	0.69	0.54–0.88
**Postoperative Infections**	**Adjusted Odds Ratio**	**Confidence Interval 95%**
Exposure to biological therapy	1.50	1.03–2.17
C-reactive protein	1.04	1.01–1.06
Hypoalbuminemia	1.92	1.27–2.90
Surgical approach: laparotomy	2.15	1.39–3.32

## Data Availability

The data underlying this article will be shared on reasonable request to the corresponding author.

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
