# Peer review of "Impact of Biological Agents on Postsurgical Complications in Inflammatory Bowel Disease: A Multicentre Study of Geteccu"

_jcm, 2021, doi:10.3390/jcm10194402_

Round 1
Reviewer 1 Report
- Please rephrase this statement, biologics have not replaced other therapies: Since then, the treatment based on corticosteroids, immunomodulators and surgery switched to biological therapies.
- Please remove extra spaces between words.
- What surgery related factors? Perhaps consider mentioning surgical procedure related risks.
- Please format the manuscript in the standard recommended format: Introduction should be followed by Methods and study design which then should be followed by Results then discussion. Refer to ‘Research Manuscript Sections’ on the journals webpage: https://www.mdpi.com/journal/jcm/instructions.
- Please replace registered with performed.
- Replace ‘at any point of the disease’ to ‘at any point during the disease course’.
- Rephrase as: ……..more than three complications.
- Table 1, please explain on what evisceration means in the results section.
- Please rephrase as: …. with lower median age at the time of surgery, …..
- Please rephrase, what does regardless of hospitalizations mean here. The statement sounds contradictory.
- I would recommend that Table 2 should be mentioned first in the results (make it table 1) then followed by Table 1 (make it table 2). That way readers can first understand the demographics of the patients that are being studied.
- It would be interesting to see if the authors can add individual p-values for CD and UC separately to see if there was a baseline difference between the groups in terms of type of IBD.
- In the current table 2: Consider writing Behavior of CD at surgery instead of behavior of IBD, if all the patients in question here are CD patients.
- In the current table 2, replace lymphocytes with Lymphocyte count.
- In current table 2 what is the difference between nutritional support and pre-operative nutrition? If they are the same and preoperative nutrition is a simple breakdown of nutritional support then avoid separating the two rows and adding two separate p-values, leads to confusion.
- ……. cohort resulting in an increase use of blood transfusions and intravenous iron in that group.
- ……cholesterol, albumin and prealbumin levels were observed….
- Please rephrase as: The factors associated with patients experiencing more postoperative complications as determined in the univariate analysis were male gender, age over 40 years at the time of surgery, diagnosis of UC, severe anaemia, corticosteroid use, higher levels of C-reactive protein (CRP) and nutritional parameters such as low serum cholesterol and albumin levels during the preoperative period (table 3).
- In Table 3 please add p-values for all categories and sub-categories to get a clear picture of any form of statistical difference between two groups. For example: under the age category, please add a p-value for age <40, 40-60 and >60; instead of mentioning one universal p-value for this category. Same for type of IBD or extent of disease.
- Please briefly describe the findings of table 4, which are quite interesting under the results section.
- Please remove the word ‘final’.
- This statement is true except for some risk of post-op infections.
- Rephrase as: ……anti-TNF (261 exposed to adalimumab and exposed to infliximab)….
- Figure 1: please label each image as A, B and C respectively.
- Figure 1: Very interesting finding of vedolizumab causing higher rates of complications and infections.
- Using the term genetic score match analysis may not fully represent the type of analysis performed. It may be termed as sensitivity analysis, but no genetic testing was performed to call it genetic sensitivity analysis. This does fall under a form of phenotypic matched analysis. Using genetic match analysis may sound misleading to the readers.
- Please rephrase this statement.
- Please remove the extra space.
- ….the complete withdrawal of biological therapy during the preoperative period is not necessary…
- Please rephrase.
- Are we clear there was no statistical difference? Since based on figure 1 there appears to be some difference especially for use of vedolizumab in CD.
- Please remove the extra space.
- Change to: …only 3 UC patients were treated with ustekinumab…..
- Change to: …..to avoid postoperative complications.
- Change to: ……revealed a slightly higher incidence of infections….
- Please remove extra space.
- anti-TNF therapy
- anti-TNF agents
- Please remove extra space.
- Change to: …. to a higher incidence of superficial wound infections, whereas….
- Change to: ….its use was associated with….
- What does bursting pressure of healing refer to? Please use adequately descriptive terminologies.
- Please rephrase.
- Please change to: However, the Clavien-Dindo classification, which has been used as an outcome in previous reports, was used to avoid this limitation thereby standardising our data.
Please make sure the genetic matched score is a term you can use here and is not misleading to the readers since no obvious gene testing was performed and the variables included in that scoring system are phenotypic ones.
- Please rephase as: Pregnant women, patients on immunosuppressants for diseases other than IBD, patients on biologicals for diseases other than IBD and patients who underwent surgeries for perianal disease were excluded.
- Please state the type of biologics under study in the methods section, also state if biosimilars were included, if so, then please state which biosimilars were included.
- Replace sorted with distributed
- Replace behavior with severity.
- Replace gender with sex
- Please delete that last sentence giving details about REDCap.
- Replace gender with sex.
- Please rephrase as “Low cholesterol levels: serum cholesterol level below 160 mg/dL at any point during the 2 weeks prior to surgery.”
- Please rephrase as “Smoking habit: current smokers included individuals who actively smoked more than 7 cigarettes per week, former smokers included individuals who quit smoking more than 6 months ago and non- smokers included those patients who had never smoked before.”
- The unit of BMI is kg/m2
- Please rephrase.
Reviewer 2 Report
I read the manuscript from the Young Group of GETECCU with high interest. Although I believe it is quite hard to define the independent factors for PC, with a large cohort it is interesting to see the outcome. The manuscript is well prepared and presented, therefore I only have minor comments for improvement;
- I recommend authors adjusting all the figures and tables as they can be 100% clear and understandable without even reading the text. To achieve this, simply provide all the necessary information (the statistical test used, the explanation on what p vallue stands for, al the abbreviations etc.) to read the table as a table explanation under.
- I recommend authors to use simply 2 decimals after comma for all p values as standard and if they want to clarify they should only use p<0.05 or p<0.001 the others like <0.0003 or <0.0001 are not usual.
- I recommend authors using * in the tables when the p-value is significant. they can use * for p<0.05 and ** for p<0.001.
- I recommend authors discussing the significant differences in clinical characteristics between the exposed and non-exposed groups and how does it affect the study outcomes.
